# Machine Learning Based Multi-Parameter Modeling for Prediction of Post-Inflammatory Lung Changes

**DOI:** 10.3390/diagnostics15060783

**Published:** 2025-03-20

**Authors:** Gerlig Widmann, Anna Katharina Luger, Thomas Sonnweber, Christoph Schwabl, Katharina Cima, Anna Katharina Gerstner, Alex Pizzini, Sabina Sahanic, Anna Boehm, Maxmilian Coen, Ewald Wöll, Günter Weiss, Rudolf Kirchmair, Leonhard Gruber, Gudrun M. Feuchtner, Ivan Tancevski, Judith Löffler-Ragg, Piotr Tymoszuk

**Affiliations:** 1Department of Radiology, Medical University Innsbruck, Anichstrasse 35, 6020 Innsbruck, Austria; anna.luger@i-med.ac.at (A.K.L.); christoph.schwabl@i-med.ac.at (C.S.); anna-katharina.gerstner@i-med.ac.at (A.K.G.); leonhard.gruber@i-med.ac.at (L.G.); gudrun.feuchtner@i-med.ac.at (G.M.F.); 2Department of Internal Medicine II, Medical University Innsbruck, Anichstrasse 35, 6020 Innsbruck, Austria; thomas.sonnweber@i-med.ac.at (T.S.); alex.pizzini@i-med.ac.at (A.P.); sabina.sahanic@i-med.ac.at (S.S.); maximilian.coen@i-med.ac.at (M.C.); guenter.weiss@i-med.ac.at (G.W.); rudolf.kirchmair@i-med.ac.at (R.K.); ivan.tancevski@i-med.ac.at (I.T.); judith.loeffler@i-med.ac.at (J.L.-R.); 3Department of Pneumology, LKH Hochzirl—Natters, In der Stille 20, 6161 Natters, Austria; katharina.cima@tirol-klinken.at (K.C.); anna.boehm@tirol-kliniken.at (A.B.); 4Department of Internal Medicine, St. Vinzenz Hospital, Sanatoriumstraße 43, 6511 Zams, Austria; ewald.woell@krankenhaus-zams.at; 5Institute of Clinical Epidemiology, Public Health, Health Economics, Medical Statistics and Informatics, Medical University of Innsbruck, Anichstraße 35, 6020 Innsbruck, Austria; piotr.tymoszuk@i-med.ac.at

**Keywords:** artificial intelligence, lung CT, quantification, COVID-19

## Abstract

**Objectives**: Prediction of lung function deficits following pulmonary infection is challenging and suffers from inaccuracy. We sought to develop machine-learning models for prediction of post-inflammatory lung changes based on COVID-19 recovery data. **Methods**: In the prospective CovILD study (*n* = 420 longitudinal observations from *n* = 140 COVID-19 survivors), data on lung function testing (LFT), chest CT including severity scoring by a human radiologist and density measurement by artificial intelligence, demography, and persistent symptoms were collected. This information was used to develop models of numeric readouts and abnormalities of LFT with four machine learning algorithms (Random Forest, gradient boosted machines, neural network, and support vector machines). **Results**: Reduced DLCO (diffusion capacity for carbon monoxide <80% of reference) was found in 94 (22%) observations. Those observations were modeled with a cross-validated accuracy of 82–85%, AUC of 0.87–0.9, and Cohen’s κ of 0.45–0.5. No reliable models could be established for FEV1 or FVC. For DLCO as a continuous variable, three machine learning algorithms yielded meaningful models with cross-validated mean absolute errors of 11.6–12.5% and R^2^ of 0.26–0.34. CT-derived features such as opacity, high opacity, and CT severity score were among the most influential predictors of DLCO impairment. **Conclusions**: Multi-parameter machine learning trained with demographic, clinical, and artificial intelligence chest CT data reliably and reproducibly predicts LFT deficits and outperforms single markers of lung pathology and human radiologist’s assessment. It may improve diagnostic and foster personalized treatment.

## 1. Introduction

Chest computed tomography (CT) has emerged as a cornerstone in the assessment of diverse lung pathologies, ranging from early-stage malignancies to chronic obstructive pulmonary disease (COPD), interstitial lung diseases (ILDs), and infection, such as SARS-CoV-2 pneumonia (COVID-19). Traditional approaches of evaluation of pulmonary CT scans have predominantly relied on manual assessment by radiologists, a process characterized by subjectivity, time intensity, and inter-observer variability [1]. The recent integration of software-based CT quantification techniques has offered a multifaceted arsenal of advantages over conventional scoring techniques, allowing for rapid and consistent analysis of voluminous imaging datasets [2]. Recent studies have demonstrated the potential of artificial intelligence (AI) to accurately detect and quantify inflammatory lung changes on CT scans [3,4]. As such, CT quantification may not only facilitate risk stratification but may also contribute to our understanding of disease progression and treatment response [5]. Furthermore, it may foster the identification of novel imaging biomarkers indicative of disease severity, prognosis, and therapeutic efficacy [6].

However, the prediction of lung function deficits following COVID-19 suffers from low accuracy [7]. Additionally, relevance of residual structural lung lesions in CT and severity of lung damage in COVD-19 convalescents for pulmonary function is not entirely clear [7,8,9].

Machine learning based multi-parameter modeling may overcome limitations of univariate biomarker screening with statistical tests, correlations, and receiver operating curve (ROC) analysis and multi-parameter modeling with human-selected explanatory factors, especially in multi-dimensional, inter-disciplinary data sets.

The machine learning approach has already been applied to predictions of outcomes and recovery trajectories in COVID-19. For instance, using support vector machines (SVM) algorithm with demographic and clinical predictors, Cordelli and colleagues predicted lung lesions one year after COVID-19 with cross-validated 94% accuracy and an area under the ROC curve (AUC) of 98% [10].

Similarly, prediction of fibrotic pulmonary lesions using XGBoost algorithm and a combination of clinical and demographic predictors, Ribeiro Carvalho et al. reported a test subset (hold-out) accuracy of 78% and AUC of 0.83 [11].

Reports by Boulogne at al. and by Park et al. demonstrate feasibility of neural networks for prediction of lung function testing (LFT) parameters with raw CT images [12,13]. Yet, although the deep learning is undoubtedly attractive for diagnostics, it is poorly explainable and does not allow for identification of demographic and clinical risk factors associated with poor lung function.

Herein, we applied four machine learning algorithms to model parameters and deficits of LFT in COVID-19 convalescents with demographic, clinical, laboratory and lung CT information. Of note, the CT data consisted of both human- and AI-derived quantification of lung damage severity. By an analysis of the importance of the explanatory variables for machine learning predictions, we sought to identify markers of impaired pulmonary function after COVID-19.

## 2. Materials and Methods

### 2.1. Study Data

Data of clinical and cardiopulmonary recovery recorded in the prospective multicenter CovILD study is described in more detail in our previous studies on the same cohort [7,8,9,14] (Medical University of Innsbruck, Austria approval number: 1103/2020) and in Appendix A. COVID-19 survivors were recruited between March and June 2020 at three clinical centers in Tyrol, Austria (*n* = 145), and were investigated at two, three, six, and twelve months after diagnosis. The study inclusion criteria were age ≥ 18 years, SARS-CoV-2 positivity confirmed by PCR and presence of typical COVID-19 symptoms. All participants were infected with the wild-type form of SARS-CoV-2. Herein, *n* = 420 longitudinal observations from 140 participants were analyzed, with complete CT of the chest and lung function testing (LFT) as analysis inclusion criteria (Figure 1, Appendix A). Please note that per participant, up to four observations were available. Due to this participant-matching, the observations were not independent, which had consequences for our analysis strategy. The dropout rates as compared with the initially recruited *n* = 145 patients were 17%, 14%, 41%, and 37% at the two-, three-, six-, and twelve-month follow-up (Appendix A).

The severity of lung lesions in CT was scored by thoracic radiologists with the previously described CT severity score (CTSS) [8]. CTSS is a semi-quantitative scoring system of overall extent and severity of structural lung abnormalities based on radiologist’s interpretation. Lesions in each lobe were graded on a 0–5-point scale, with 0 corresponding to no abnormalities and 5 corresponding to extensive parenchymal destruction. The CTSS was calculated as the sum of all five lobes (maximum score: 25 points). In addition, an artificial intelligence-based software (Syngo.via CT Pneumonia Analysis Software, version 2; Siemens Healthineers, Erlangen, Germany) was used. It is an AI-based, quantitative tool that objectively measures opacity, which reflects all dense lung alterations, primarily ground-glass opacities (GGO), and high opacity, which corresponds to consolidation, based on CT attenuation values [9,14].

### 2.2. Analysis Outcomes and Endpoints

The following numeric LFT parameters were analyzed as percentage of patient’s reference values: DLCO (diffusion capacity for carbon monoxide), FVC (forced vital capacity), and FEV1 (forced expiratory volume in the first one second). Functional lung abnormalities were defined as hemoglobin corrected DLCO < 80%, FVC < 80%, and FEV1 < 80% of the patient’s reference.

The primary analysis endpoint was construction and evaluation of multi-parameter models of the LFT abnormalities (each of DLCO < 80%, FVC < 80%, FEV1 < 80% of the patient’s reference) and of numeric LFT readouts (DLCO, FVC, FEV1) during COVID-19 convalescence with baseline and longitudinal demographic, clinical, and CT variables as explanatory factors. The primary endpoint was addressed by a machine learning modeling approach.

The secondary analysis endpoints were an analysis of the importance of the explanatory variables for the machine learning LFT predictions and assessment of human-determined CTSS and AI-measured CT lung opacity and high opacity as standalone predictors of LFT abnormalities and numeric LFT readouts. These endpoints were addressed by Shapley additive explanations (SHAP) as well as statistical hypothesis testing, correlation, and ROC analysis.

### 2.3. Statistical Analysis

Details of statistical analysis are provided in Appendix A.

Statistical analysis was performed with R version 4.2.3 (R Foundation for Statistical Computing). Numeric variables were presented as medians with interquartile ranges and ranges. Qualitative variables were presented as percentages and counts of the categories within the complete observations set. Differences in independently distributed numeric variables were analyzed by Mann–Whitney and Kruskal–Wallis test with, respectively, biserial r and η^2^ effect size statistic. Statistical significance for differences in distribution of qualitative variables was determined by χ^2^ test with Cramer’s V effect size statistic. Co-occurrence of each of LFT findings, CT abnormalities, and symptoms were investigated by two-dimensional correspondence analysis.

Differences in medians of non-independently distributed, participant matched numeric variables between observations with and without LFT and CT abnormalities were assessed by a blocked bootstrap test with blocks defined by the participant’s identifier, and effect size measured by biserial r effect size statistic. Correlations of non-independently distributed CT and LFT readouts were assessed by blocked bootstrap Spearman’s rank test. Cutoffs of CTSS, opacity and high opacity for detection of LFT abnormalities were found by maximizing the Youden’s J statistics. ROC analysis statistics (area under the curve [AUC], sensitivity, specificity, Cohen’s κ) for those optimal cutoffs were computed and their 95% confidence intervals were obtained by blocked bootstrap.

Reduced DLCO, FVC, and FEV1 (each < 80% of reference), as well as values of DLCO, FVC, and FEV1 expressed as percentages of the patient’s reference were modeled with 37 explanatory variables. The explanatory variables included demographic features (e.g., age, sex, body mass index, smoking, comorbidity), characteristic of acute COVID-19 (severity, medication) and recovery (e.g., weight loss, symptoms of relevance for lung function, time after diagnosis), and presence of human- and AI-rated structural lung abnormalities in CT scans (e.g., GGO, CTSS, opacity, and high opacity). The modeling responses and explanatory variables are listed in Appendix A. The models were constructed with four machine learning algorithms: Random Forest, gradient boosted machines (GBM), neural network with a single hidden layer, and support vector machines (SVM) with radial kernel. Selection of the optimal values of parameters controlling model behavior such as number of random trees, neurons in the hidden layer, or cost parameter was motivated by the maximum of Youden’s J statistic (classification models of LFT abnormalities) or minimum mean absolute error (MAE, regression models of LFT readouts) in 10-repeats 10-fold cross-validation [15,16,17]. Because of the presence of participant-matched observation, blocked cross-validation design was used both in the model selection and model evaluation, with blocks defined by participant’s identifier. Model predictions were evaluated both in the training data and blocked 10-repeats 10-fold cross-validation. Concordance between the predicted model and observed outcomes for classification models was assessed by Cohen’s κ inter-rater reliability statistic. Accuracy, AUC, specificity, and sensitivity of the classification model were investigated by ROC. Calibration of the classification models was assessed by Brier scores. Fraction of explained variance in predictions of the regression models was measured by pseudo-R^2^, the regression model error was expressed as MAE. Spearman’s ρ coefficients of correlation between the predicted and observed values were used to gauge calibration of the regression models. Over- and under-fitting was assessed by learning curves. Importance of explanatory variables was estimated by absolute values of SHAP statistics (Shapley additive explanations). Co-linearity of the most influential explanatory variables (top 15 mean absolute SHAP for each of the models of DLCO and DLCO < 80%) was assessed by soft-threshold weighted graph of correlations. The graph edges were defined by pairwise correlations with Kendall’s τ ≥ 0.3 and edge weights corresponded to τ coefficient values.

## 3. Results

### 3.1. Machine Learning Prediction of Post Inflammatory Lung Function Impairment

Insufficient DLCO (22% of observations, *n* = 94), FVC (20%, *n* = 83), and FEV1 (18%, *n* = 77) defined as values below < 80% of the patient’s reference value were the most common abnormalities of lung function (Appendix A, Appendix A).

Among the lung function test (LFT) parameters (Appendix A, Appendix A), only reduced DLCO < 80% yielded meaningful models with reproducible accuracy between the algorithms as evaluated by cross-validation (overall accuracy: 82–85%, κ: 0.45–0.5, AUC: 0.87–0.9). The models also showed good calibration, as indicated by low Brier scores (0.11–0.14). In contrast, models of reduced FVC and FEV1 performed poorly (accuracy: 72–81%, κ: 0.094–0.17, AUC: 0.57–0.69). Similar trends were observed for models of continuous DLCO, where Random Forest, GBM, and SVM algorithms demonstrated the best performance (cross-validation, MAE: 11.6–12.5, pseudo-R^2^: 0.26–0.34) and strong correlation between predicted and observed DLCO (Spearman’s ρ: 0.55–0.59). The neural network model, however, performed poorly in cross-validation (MAE: 13.8, pseudo-R^2^: 0.043) (Figure 2B). No meaningful models could be developed for FVC or FEV1 (R^2^: −0.086 to −0.03) (Figure 2, Table 1 and Table 2, Appendix A, Appendix A).

The models for DLCO < 80% performed best in moderate COVID-19 survivors at 2–6 months post-infection (all algorithms, κ: 0.45–0.69). Performance was the poorest for ambulatory patients at 6–12 months follow-up (κ: 0–0.46). Random Forest, SVM, and GBM models of continuous DLCO showed the lowest errors for moderate COVID-19 (mean error: −2.8 to 2.9). However, DLCO was systematically overestimated in severe COVID-19 (mean error: 1.2 to 6.5) and underestimated in ambulatory cases (mean error: −4.2 to 0.12, Appendix A). The better performance in moderate cases was likely due to the larger number of observations and frequent DLCO impairments in this group (*n* = 47 with DLCO < 80% out of 234 observations with moderate COVID-19), whereas fewer data points from ambulatory patients led to reduced model accuracy.

Prediction of reduced DLCO < 80% was assessed in observations with increasing values of DLCO expressed as a percentage of the patient’s reference by moving averages of accuracy and squared distance to the 0/1-coded outcome. For the best performing GBM model (Appendix A) and the remaining models, the accuracy was the lowest and the distance to the outcome peaked for DLCO between 70% and 80%. This illustrates that while predictions of highly compromised and normal-to-high DLCO are accurate, forecasts for borderline reduced DLCO suffer from error. This may question the biological and clinical relevance of the 80% cutoff of insufficient DLCO in our COVID-19 cohort.

To assess under- and over-fitting, we resorted to analyses of learning curves of the models of DLCO and reduced DLCO re-trained in subsets of the modeling data set of varying sizes. The performance evaluation was performed for the training subsets, test subsets (one-fourth of observations not used for the model training), and 10-repeats 10-fold cross-validation (21). As inferred from substantial differences in accuracy and Cohen’s κ between the training, test, and cross-validation subsets even for the largest sizes of the training data, models of insufficient DLCO suffer from over-fitting, i.e., poor generalizability for unseen data. Convergence of performance trajectories in the training, test, and cross-validation subsets for the models of continuous DLCO speak for a substantially better generalizability (representative for the GBM model: Appendix A).

### 3.2. Key Predictors of DLCO

As investigated by absolute values of SHAP [18], human- and AI-derived ratings of structural lung damage (CTSS, opacity and high opacity, GGO, reticulation, bronchiectasis), risk factors of severe COVID-19 (age, male sex, body mass index, co-morbidity), readouts of severe acute infection (severity class, hospitalization and ICU stay, anti-coagulant and anti-infective treatment, weight change), smoking and impaired physical performance belonged to the most influential explanatory variables for predictions of DLCO and DLCO insufficiency (Figure 3, Appendix A, Appendix A). In particular, the highly influential CT-related variables were inter-correlated, which raises the question about their redundancy (Appendix A, Appendix A). Of note, controlling this redundancy, e.g., by regularized machine learning algorithms may further improve accuracy of the DLCO predictions.

### 3.3. CT Markers of Lung Function Impairment

CT features alone were significant indicators of DLCO impairment. Human-assessed CTSS (median difference: 9 points, *p* < 0.001, effect size: r = 0.57), software-derived opacity (Δ: 1.3% of lung volume, *p* < 0.001, r = 0.63), and high-opacity regions (Δ: 0.063% of lung volume, *p* < 0.001, r = 0.58) were all significantly elevated in cases with reduced DLCO (Figure 4, Appendix A).

In ROC analysis, software-derived opacity (cutoff: 0.12% of lung volume) demonstrated the best performance for identifying reduced DLCO (AUC: 0.81, sensitivity: 0.81, specificity: 0.69). High opacity (cutoff: 0.002%, AUC: 0.79) and CTSS (cutoff: 4 points, AUC: 0.78) were slightly less effective but still relevant markers (Figure 4, Table 3). The cutoff values for opacity and high opacity were very small, but, as presented in Figure 5 for a representative CovILD study participant, opacity in <2% of the lung were already associated with severe DLCO insufficiency.

## 4. Discussion

We demonstrate that machine learning fed with a combination of demographic, clinical, and radiological data can effectively predict DLCO impairment following COVID-19, particularly in moderate disease cases. The superior accuracy of these models (cross-validated AUC: 0.87–0.90) compared to standalone CT-based markers such as CTSS, opacity, or high opacity (AUC: 0.78–0.81) underscores the multifactorial nature of post-COVID lung function impairment, which is influenced not only by structural lung changes but also by clinical and demographic factors.

A study by Ma et al. similarly developed a machine learning model for predicting DLCO impairment in COVID-19 survivors using clinical and laboratory data [19]. Their XGBoost model achieved an AUC of 0.76 and an accuracy of 78%, slightly lower than the performance of our models (AUC: 0.87–0.90, accuracy: 82–85%). While Ma et al. identified hemoglobin levels, maximal voluntary ventilation (MVV), platelet count, uric acid, and blood urea nitrogen as the most influential predictors, our models relied more heavily on CT-derived markers (opacity, high opacity, CTSS) in addition to known risk factors of severe COVID-19 and readouts of severe acute infection. This difference highlights the potential complementary value of integrating both structural CT imaging and physiological biomarkers to enhance prediction accuracy.

Furthermore, the study by Savushkina et al. aimed to predict DLCO impairment using a statistical model based on semi-quantitative CT evaluation of lung abnormalities during the acute phase of COVID-19 [20]. Instead of applying artificial intelligence-based quantification, their approach relied on human-derived CT severity scoring, like our CTSS. Their logistic regression model showed that severe lung involvement on CT (≥45%) was significantly associated with reduced DLCO (OR: 1.21, AUC: 0.78). While this aligns with our findings that CT-based features are strong predictors, our model improves upon their approach by integrating a broader set of explanatory variables, including laboratory biomarkers, clinical history, and demographic factors, leading to a higher predictive accuracy.

Several other research groups have also employed machine learning approaches to predict pulmonary function outcomes based on CT imaging. Boulogne et al. developed a deep learning model to estimate DLCO, FEV1, and FVC directly from CT scans at both the patient and lobe levels, achieving a mean absolute error (MAE) of 2.8 mL/min/mmHg for DLCO prediction [12]. Their study demonstrated that machine learning can extract functional information from CT scans beyond traditional radiological assessment. Similarly, Park et al. applied a deep learning-based approach to predict FEV1 and FVC from low-dose chest CT scans, reporting a strong correlation with spirometry-derived values (concordance correlation coefficient: 0.94 for FVC, 0.91 for FEV1) [13]. While these studies successfully linked CT-derived features to lung function parameters, they did not specifically evaluate long-term post-COVID pulmonary impairment. Notably, Boulogne et al. and Park et al. utilized distinct cohorts, including healthy individuals and patients from the COPDGene study, while our study focused specifically on COVID-19 survivors. Despite these differences, all models demonstrated the feasibility of predicting pulmonary function from CT scans, highlighting that machine-learning-based lung function estimation from CT may be applicable across different pulmonary conditions. This suggests that in the future, independent of the underlying lung disease, machine-learning models could enable the estimation of pulmonary function parameters directly from chest CT scans.

In our study, both scoring systems (human-derived CTSS and AI-based quantification) contribute to DLCO prediction. This highlights the synergy between human and artificial intelligence, showing that automated CT imaging analysis can complement radiologist assessments. Compared with human scoring, artificial intelligence can detect and quantify lung changes automatically and independently from a reader’s experience or inter-reader agreement [8]. Automated quantification may also facilitate comparison of serial CT scans [8,21]. Human scoring is time consuming, and defining disease stages related to visual estimation of percentages of involved lung zones or separate quantification of different patterns, such as ground-glass, and consolidation can be challenging [22,23,24]. Humans may use software for segmentation, but even semi-automated segmentation is time consuming [25]. As a drawback, artificial intelligence may misinterpret artifacts from respiratory motion. Most artificial intelligence software tools quantify lung involvement based on CT density only. The full spectrum of lung pathology (e.g., reticulation, GGO, bronchial dilation, atelectasis, cystic lesions, vascular abnormalities, and many more) is currently not available.

In our study, artificial intelligence derived opacity of the lung, but not CTSS or high opacity, was found to be significantly higher in observations with reduced FVC and insufficient FEV1. Interestingly, the CT readouts of structural lung damage also correlated significantly with FVC and FEV1 (Appendix A). FEV1 and FVC alone are not very sensitive for detecting mild to moderate interstitial changes (e.g., mild GGO). Associations with bronchial changes may contribute to this effect. CTSS, opacity, and high opacity correlated negatively with moderate effect size with DLCO (Appendix A). DLCO is sensitive but non-specific and is usually subject to quite strong fluctuations in the longitudinal course. The comorbidities/risk factors of severe acute COVID-19 probably also have a significant influence here (age, male sex, body mass index, pre-existing malignancy and cardiovascular disease, severity class, WHO ordinal severity scale, hospitalization length, ICU treatment, anti-coagulant treatment, weight change, smoking intensity, rating of physical performance impairment, and exertional dyspnea) [26,27].

Interestingly, our ROC analysis revealed extremely low cutoff values for opacity (0.12%) and high opacity (0.002%) when used as standalone markers of DLCO impairment. Similar findings were reported by Compagnone et al., who observed a high prevalence of both structural and functional lung deficits in COVID-19 ARDS survivors [28]. Sixty percent of patients exhibited reduced DLCO, consistent with persistent gas exchange impairment. However, despite this high prevalence of DLCO reduction, the structural lung changes were relatively mild, like those observed in our study. This suggests that even minimal residual lung abnormalities could have significant clinical implications, reinforcing the need for careful long-term monitoring of COVID-19 survivors with persistent symptoms.

Prediction models were found to have a high risk of bias, related to non-representative selection of control patients, exclusion of patients who had not experienced the event of interest by the end of the study, and model overfitting [5]. Nevertheless, future studies should focus on refined machine learning models based on the full spectrum of longitudinal clinical and imaging data to improve current prediction models [21,29,30,31,32].

### Limitations

Our study has several limitations. First, the overall patient and observation number was low, in particular for ambulatory COVID-19 convalescents. Analogically, the study cohort was enriched in hospitalized individuals which constitute a minute fraction of COVID-19 patients in the real-world setting. Second, complete sets of longitudinal CT and LFT measurements at the two-, three-, six-, and twelve-month follow-up examinations were available solely for 55 patients. In particular, the number of observations obtained with ambulatory and moderate COVID-19 convalescents at the six- and twelve-month follow-ups were substantially lower as compared with earlier time points. The incompleteness of the longitudinal data may have compromised the performance of the machine learning models, in particular for ambulatory COVID-19 cases and at the later time points. Third, because of the limited number of participants and observations, we abstained from the definition of a test subset of the data used solely for bias-free model evaluation (hold-out strategy). Instead, both model selection and evaluation were performed with blocked repeated cross-validation, which may have overestimated performance of the models. Hence, external validation of our findings in an independent cohort is recommended. Fourth, as inferred from the analysis of learning curves, the models of insufficient DLCO suffered from substantial over-fitting. While this problem can be partially traced back to an unsharp distinction between sufficient and reduced DLCO with the 80% of the patient’s reference cutoff, a modeling approach employing regularized machine learning algorithm like XGBoost or regularized neural networks may further improve quality of predictions. Fifth, most of the highly influential explanatory variables for predictions of DLCO and reduced DLCO, and, in particular, the CT readouts of lung damage were strongly inter-correlated, which raises questions about redundancy of explanatory variables. Furthermore, CT-related explanatory variables were already processed and available in the form of software- and human-derived severity metrics and findings without any spatial information. The inclusion of raw CT image information in the machine learning models would likely enhance their accuracy, as demonstrated by literature reports. Finally, the analyzed cohort was recruited in the initial phase of the pandemic and consisted of individuals infected with the wild-type variant of the SARS-CoV-2 virus. For this reason, it is not completely clear how our findings translate to the recent variants of the pathogen and how the pulmonary recovery is affected by anti-SARS-CoV-2 immunity, improved treatment, and care. However, it is feasible that the CT severity readouts, human-determined CTSS as well as AI-determined opacity and high opacity, are equally applicable in the post-pandemic setting as standalone markers of functional lung impairment during recovery from COVID-19 and other respiratory infections.

## 5. Conclusions

This study demonstrates the feasibility of using machine-learning-based models to predict DLCO impairment in COVID-19 survivors by integrating CT-derived markers, demographic data, and clinical parameters from both the acute infection phase and early convalescence. Machine learning outperformed univariable correlations and ROC analyses, highlighting its potential for more accurate risk assessment.

The key practical implications of our findings include the early identification of COVID-19 survivors at risk for persistent lung function impairment (DLCO < 80%), allowing for targeted follow-up and timely intervention. Additionally, our results underscore the complementary value of AI-driven CT quantification as an adjunct to radiologist assessment, supporting more objective and standardized evaluations in post-COVID lung disease management.

## Figures and Tables

**Figure 1 diagnostics-15-00783-f001:**
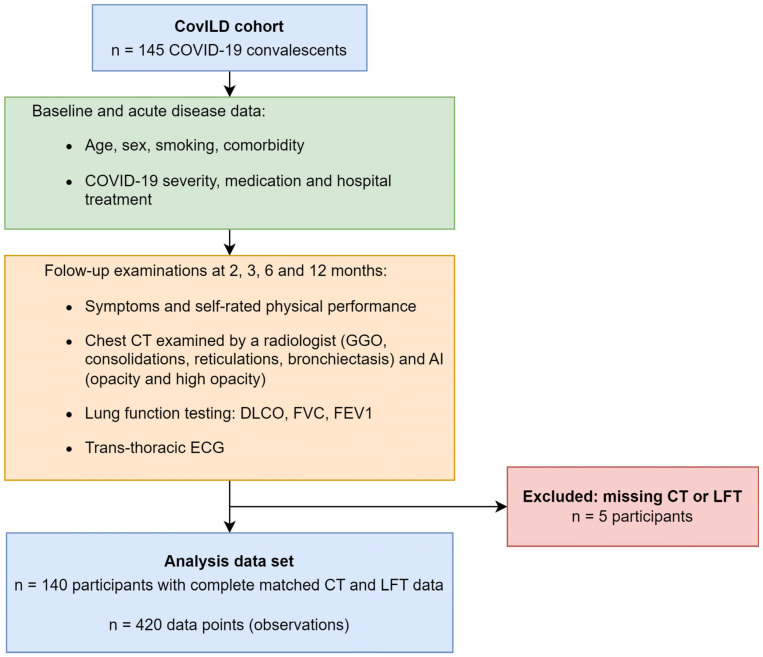
**Analysis inclusion scheme.** The analysis inclusion criterion for participants of the longitudinal observation CovILD study was completeness of visit-matched computed tomography and lung function testing results. *Abbreviations*: ECG: electrocardiogram; CT: computed tomography; LFT: lung function testing; GGO: ground-glass opacity; AI: artificial intelligence; DLCO: diffusion capacity for carbon monoxide (CO); FVC: forced vital capacity; FEV1: forced expiratory volume in one second.

**Figure 2 diagnostics-15-00783-f002:**
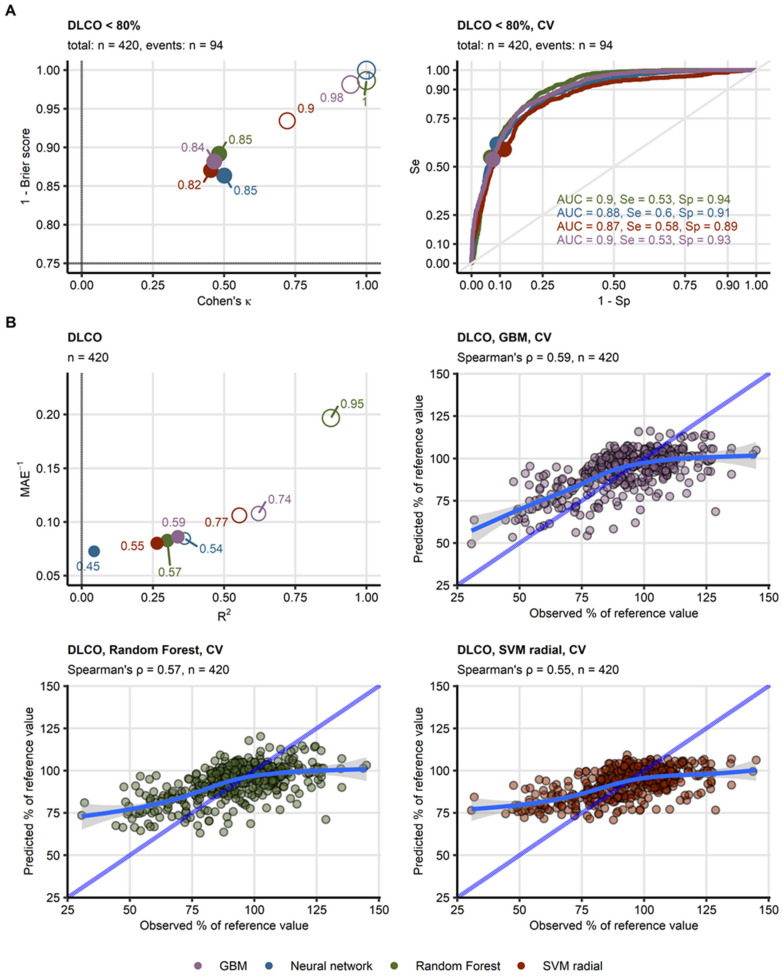
**Evaluation of performance of machine learning models of diffusion capacity for carbon monoxide during COVID-19 convalescence.** (**A**) Four machine learning classification models of insufficient diffusion capacity for carbon monoxide (<80% of reference value: *n* = 94, total observations: *n* = 420) employing time after COVID-19 diagnosis, computed tomography readouts, demographic and clinical explanatory variables were trained. Their performance was evaluated in the entire data set and 10-repeats 10-fold cross-validation with overall accuracy metric, Cohen’s κ as a measure of concordance between predicted and observed outcome, and Brier score as a measure of model’s calibration. Left: numeric performance measures of the models (open circles: the entire data set, filled circles: cross-validation); point sizes and point labels represent overall model accuracy, the dashed lines visualize values of Cohen’s κ and Brier score expected for prediction of insufficient DLCO be chance. Right: receiver-operating characteristic curves for predictions in cross-validation folds, numeric statistics are displayed in the plot. Numbers of complete observations and observations with DLCO insufficiency (events) are displayed in the plot captions. (**B**) Four machine learning regression models of diffusion capacity for carbon monoxide (percentage of reference values, total observations: *n* = 420) employing time after COVID-19 diagnosis, computed tomography readouts, demographic and clinical explanatory variables were trained. Their performance was evaluated in the entire data set and 10-repeats 10-fold cross-validation with R^2^ as a measure of explained variation, mean absolute error, and ρ Spearman’s coefficient of correlation between the predicted and observed values. Bubble plot: numeric performance measures of the model (open circles: the entire data set, filled circles: cross-validation); point sizes and point labels represent values of ρ correlation coefficient, the dashed line visualizes R^2^ value expected for a meaningless model. Scatter plots: observed and predicted values of diffusion capacity for carbon monoxide in cross-validation folds; the blue dashed lines with slope 1 and intercept 0 represent absolutely accurate predictions, general additive model trends with standard errors are visualized as the solid blue lines with gray ribbons. Numbers of complete observations and Spearman’s ρ coefficients of correlation between the predicted and observed values are displayed in the plot captions. Ranges of the displayed DLCO values were set to the range of observed DLCO. *Abbreviations*: DLCO: diffusion capacity for carbon monoxide, CV: cross-validation; AUC: are under the receiver-operating characteristic curve; Se: sensitivity; Sp: specificity; MAE: mean absolute error; GBM: gradient boosted machines; SVM radial: support vector machines with radial kernel.

**Figure 3 diagnostics-15-00783-f003:**
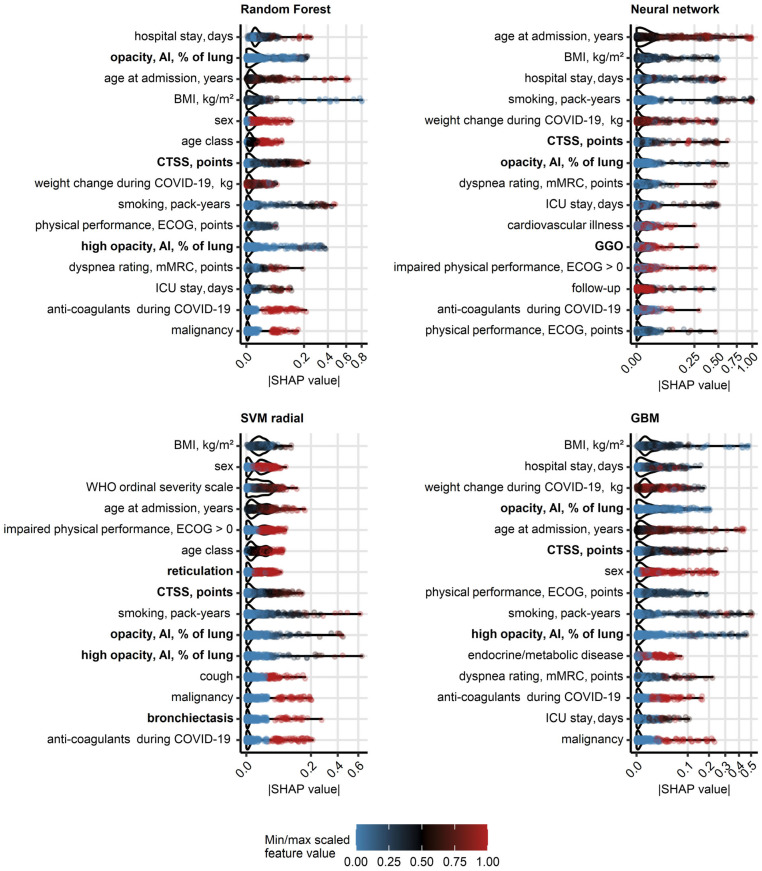
**Explanatory variable importance for models of insufficient diffusion capacity for carbon monoxide measured by Shapley additive explanations**. Importance of explanatory variables for the machine learning models of insufficient diffusion capacity for carbon monoxide (<80% of reference, Figure 2) was investigated by Shapley additive explanations (SHAP). Absolute SHAP values for explanatory variables with the 15 largest mean SHAP values are presented in violin plots. Points represent single observations, point colors code for minimum/maximum scaled value of the explanatory variable. Explanatory variables obtained via computed tomography are highlighted with bold font in the Y axes. *Abbreviations*: CT: computed tomography; DLCO: diffusion capacity for carbon monoxide; opacity and high opacity, AI: opacity and high opacity of the lung determined by artificial intelligence; BMI: body mass index; CTSS: human-determined CT severity score, sum for all lobes; ECOG: Eastern Cooperative Oncology Group physical performance score; mMRC: modified Medical Research Council dyspnea scale; ICU: intensive care unit.

**Figure 4 diagnostics-15-00783-f004:**
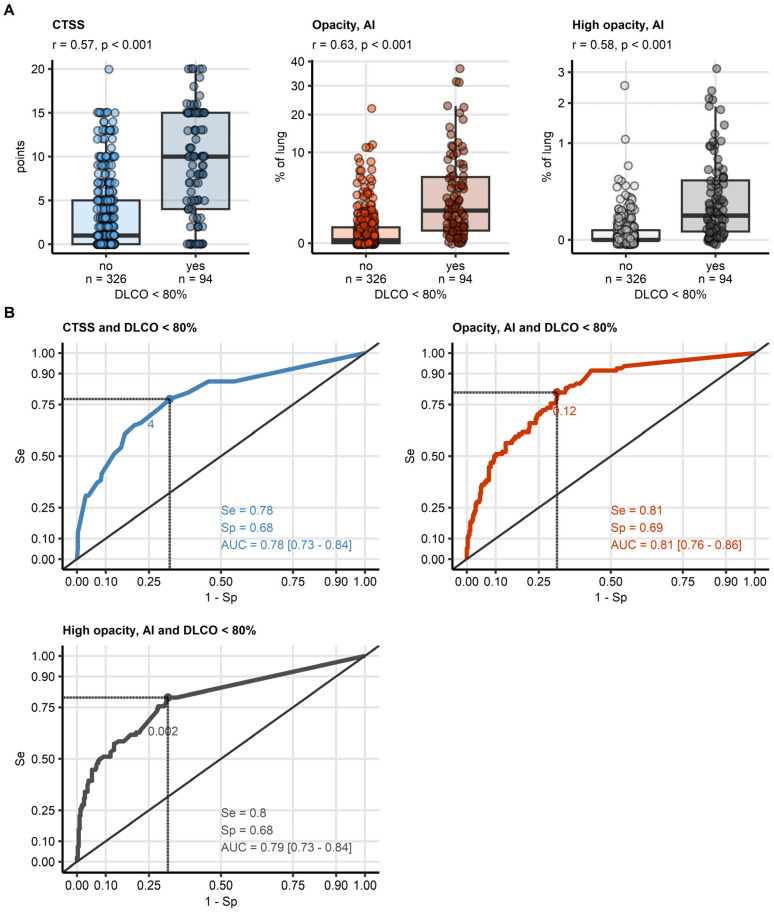
**Detection of DLCO insufficiency by human- and artificial intelligence-determined CT readouts of severity of structural lung damage.** Human- and artificial intelligence-determined computed tomography readouts of structural lung damage were identified as influential explanatory variables at prediction of insufficiency of diffusion capacity for carbon monoxide (<80%) by machine learning. (**A**) Values of the radiological readouts of lung damage severity were compared between data points with and without insufficient diffusion capacity for carbon monoxide by blocked bootstrap test with r effect size statistic. Median values with interquartile ranges are depicted as boxes, whiskers span over 150% of the interquartile ranges, single observations are visualized as points. Effect sizes and *p* values are displayed in the plot captions. Numbers of observations are indicated in the X axes. (**B**) Quality of detection of insufficient diffusion capacity for carbon monoxide with the radiological readouts of lung damage severity was assessed by receiver-operating characteristic (ROC) analysis. ROC curves are shown, the optimal cutoffs of the severity readouts determined by Youden’s criterion are represented by points with numbers. Sensitivity, specificity at the optimal cutoff, and area under the curve statistic with 95% confidence interval are displayed in the plots. *Abbreviations*: CT: computed tomography; DLCO: diffusion capacity for carbon monoxide; CTSS: human-determined CT severity score, sum for all lobes; opacity and high opacity, AI: opacity and high opacity of the lung determined by artificial intelligence; AUC: are under the curve of receiver-operating characteristic; Se: sensitivity; Sp: specificity.

**Figure 5 diagnostics-15-00783-f005:**
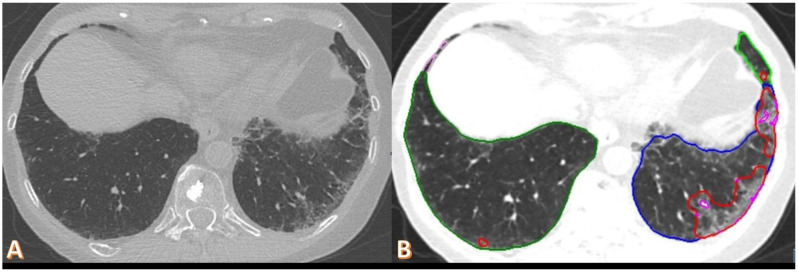
**Axial non-contrast chest CT scan of a 70-year-old female, one year after COVID-19 pneumonia.** The imaging demonstrates mild subpleural ground-glass opacities and reticulation in the left lung, and minimal involvement on the right side subpleural (**A**). Automated software quantification highlights opacity (red) and high-opacity regions (pink) within the same slice, with a measured opacity of 1.86% and high opacity of 0.06% (**B**). The patient’s DLCO was 60%.

**Table 1 diagnostics-15-00783-t001:** Cross-validated performance of binary machine learning classifiers at predicting lung function testing (LFT) abnormalities.

Response ^a^	Algorithm ^b^	Overall Accuracy ^c^	κ ^d^	Brier Score	AUC ^e^	Sensitivity	Specificity
DLCO < 80%	Random Forest	0.85	0.480	0.11	0.90	0.53	0.94
	Neural network	0.85	0.500	0.14	0.88	0.60	0.91
	SVM radial	0.82	0.450	0.13	0.87	0.58	0.89
	GBM	0.84	0.470	0.12	0.90	0.53	0.93
FVC < 80%	Random Forest	0.79	0.110	0.16	0.69	0.14	0.95
	Neural network	0.72	0.094	0.25	0.58	0.27	0.83
	SVM radial	0.78	0.120	0.16	0.68	0.19	0.92
	GBM	0.78	0.150	0.17	0.67	0.21	0.92
FEV1 < 80%	Random Forest	0.80	0.120	0.15	0.64	0.15	0.95
	Neural network	0.75	0.110	0.21	0.57	0.26	0.86
	SVM radial	0.80	0.130	0.16	0.59	0.18	0.94
	GBM	0.81	0.170	0.16	0.61	0.21	0.94

^a^ LFT: lung function testing, DLCO: diffusion capacity for CO, FVC: forced vital capacity; FEV1: forced expiratory volume in one second. ^b^ SVM: support vector machines with radial kernel; GBM: gradient boosted machines. ^c^ Ratio of correct predictions to the total observation number. ^d^ Cohen κ statistic of inter-rater reliability between the predicted and observed outcome. ^e^ AUC: are under the receiver-operating characteristic curve.

**Table 2 diagnostics-15-00783-t002:** Cross-validated performance of regression machine learning models at predicting values of lung function testing parameters.

Response ^a^	Algorithm ^b^	Pseudo-R^2 c^	MAE ^d^	ρ ^e^
DLCO	Random Forest	0.300	12	0.570
	Neural network	0.043	14	0.450
	SVM radial	0.260	12	0.550
	GBM	0.340	12	0.590
FVC	Random Forest	−0.030	10	0.220
	Neural network	−0.079	11	0.074
	SVM radial	−0.031	10	0.210
	GBM	−0.040	10	0.200
FEV1	Random Forest	−0.045	12	0.160
	Neural network	−0.086	12	0.210
	SVM radial	−0.039	11	0.190
	GBM	−0.047	12	0.170

^a^ DLCO: diffusion capacity for CO, FVC: forced vital capacity; FEV1: forced expiratory volume in one second. ^b^ SVM: support vector machines with radial kernel; GBM: gradient boosted machines. ^c^ Defined as 1—ratio of mean squared error and variance. ^d^ MAE: mean absolute error. ^e^ ρ: Spearman coefficient of correlation between the predicted and observed response values.

**Table 3 diagnostics-15-00783-t003:** Detection of reduced diffusion capacity for CO DLCO < 80% reference value) by single CT-derived parameters: AI-determined opacity and high opacity, and human-determined CT severity score.

CT Variable ^a^	Cutoff ^b^	Statistic ^c^	Value, 95% CI
CTSS		AUC	0.78 [0.727–0.84]
	4.000	κ	0.34 [0.23–0.45]
	4.000	Sensitivity	0.78 [0.64–0.89]
	4.000	Specificity	0.68 [0.6–0.75]
high opacity, AI		AUC	0.79 [0.734–0.84]
	0.002	κ	0.37 [0.26–0.47]
	0.002	Sensitivity	0.8 [0.7–0.89]
	0.002	Specificity	0.68 [0.62–0.75]
opacity, AI		AUC	0.81 [0.763–0.86]
	0.120	κ	0.38 [0.27–0.48]
	0.120	Sensitivity	0.81 [0.72–0.89]
	0.120	Specificity	0.69 [0.62–0.75]

^a^ CTSS: human-determined CT severity score, sum for al lung lobes; high opacity and opacity, AI: percentage of the lungs with high opacity and opacity determined by artificial intelligence. ^b^ Cutoff of the CT variable corresponding to the maximum of Jouden Y statistic. ^c^ AUC: area under the curve of receiver-operating characteristic; κ: Cohen κ statistic of inter-rater reliability between the predicted and observed outcome, computed for the CT variable cutoff; sensitivity and specificity: sensitivity and specificity computed at the CT variable cutoff.

## Data Availability

The original contributions presented in the study are included in the article/Appendix A, further inquiries can be directed to the corresponding author.

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
