# Peer review of "Machine Learning Based Multi-Parameter Modeling for Prediction of Post-Inflammatory Lung Changes"

_diagnostics, 2025, doi:10.3390/diagnostics15060783_

Round 1

Reviewer 1 Report

Comments and Suggestions for Authors

Dear Authors, 
Thank you for submitting the article, After carefully review the article am sharing with you the following concerns,

1. Add results in the abstract as well.  
2. The introduction is not sufficient. 
3. The literature review part needs to be added. 
4. After adding the literature review, add a summary table and clearly mention this study's research gaps and contributions. 
5. More details, such as features, are required in the study data.
6. The paper lacks details about hyperparameters for machine learning models. For example, what was the C and class weight value for the SVM model? 
7. Have you considered the GridSearchCV for model hyperparameters or set it manually? 
8. Comparison with state-of-the-art studies is missing. 
9. The Practical Implication of the study is missing. 
10. Conclusion also not well presented. 

Overall, the paper presentation regarding the write-up and experimental results is weak. 

Comments on the Quality of English Language

The English could be improved to more clearly express the research.

Author Response

Point 1

Issue

Add results in the abstract as well.

Response

As suggested, we state now in Abstract, that we were able to establish meaningful models of DLCO and DLCO < 80% but not for other LFT responses. We also included key cross-validated performance statistics of meaningful machine learning models at predicting DLCO < 80% (accuracy: 0.82 - 0.85, Cohen’s : 0.45 - 0.5, AUC: 0.87 - 0.9) and DLCO expressed as percentage of the patient’s reference (mean absolute error [MAE]: 11.6 - 12.5%, : 0.26 - 0.34). Please note that the later performance statistics of DLCO as % of the patient’s reference, we provide the performance metric range for the random forest, GBM, and SVM algorithm, as the neural network model yielded a poor fit (MAE: 14%, : 0.043).

Point 2

Issue

The introduction is not sufficient.

Response

We appreciate the reviewers’ comments regarding the Introduction. In response, we have substantially revised the Introduction to better address existing multi-parameter models for predicting post-COVID lung function, particularly studies that have used machine learning approaches.

We have also clearly highlighted the research gaps in this field and outlined how our study addresses these gaps by integrating clinical, laboratory, and CT-derived imaging features into a machine-learning framework.

Point 3 and 4

Issue

The literature review part needs to be added.

After adding the literature review, add a summary table and clearly mention this study’s research gaps and contributions.

Response

We acknowledge the reviewer’s suggestion regarding the need for a literature review. In response, we conducted an extensive literature search on machine learning applications for predicting post-COVID lung function impairment. However, we found that only one study (Ma et al., 2023) specifically applied machine learning to predict DLCO impairment using biochemical markers. The majority of previous research has focused on detection of COVID-19, prediction of mortality in acute COVID-19 and prediction of long COVID. However, to address the reviewer’s request, we have now added a brief overview of relevant studies that applied machine learning to other post-COVID outcomes. We would like to clarify that a comprehensive literature review was not the primary focus of this study, as our work is primarily centered on developing and evaluating a machine-learning-based model for predicting lung function impairment in COVID-19 survivors. Additionally, we have opted to summarize the relevant literature within the text rather than using a comparative table, as the studies identified differ substantially, making a direct tabular comparison challenging. Instead, we provide a structured discussion in the Introduction and Discussion, which we believe improves clarity and relevance.

Point 5

Issue

More details, such as features, are required in the study data.

Response

Our manuscript is accompanied by a thorough information on patients, procedures, and study variables provided in Supplementary Methods. Please refer to Supplementary Table S2 for a detailed listing of modeling responses and explanatory variables with their format and descriptions.

Point 6

Issue

The paper lacks details about hyper-parameters for machine learning models. For example, what was the C and class weight value for the SVM model?

Response

Thanks for this point. We agree that the information on hyper-parameters in the initial submission was hard to understand for many clinical readers and too specific. In the revised manuscript, we provide short descriptions of the hyper-parameters in Supplementary Methods/Multi-parameter modeling of lung function. These descriptions were also added as footnotes of Supplementary Table S6.

Point 7

Issue

Have you considered the GridSearchCV for model hyper-parameters or set it manually?

Response

Thanks for this question. The hyper-parameters were found by searching grids of values with the maximum Youden’s J statistic for the classification models, and minimum mean absolute error (MAE) for the regression models in 10-repeats 10-fold patient-blocked cross-validation as the selection criteria. This tuning procedure was executed by calling train() function from caret package (1), which, together with our open-source package caretExtra, belongs to our standard toolbox for explainable machine learning. This information is provided in Supplementary Methods/Multi-parameter modeling of lung function.

Point 8

Issue

Comparison with state-of-the-art studies is missing.

Response

We appreciate the reviewer’s request for a more detailed comparison with state-of-the-art studies. In response, we have expanded the Discussion to provide a comprehensive comparison with recent research that has applied machine learning for pulmonary function prediction.

Point 9

Issue

The Practical Implication of the study is missing.

Response

We value the reviewer’s feeback. In response, we have expanded the Discussion sections to explicitly outline how our findings can be translated into clinical practice (see page 13, end of discussion section).

Point 10

Issue

Conclusion also not well presented.

Response

In response to the reviewer’s suggestion, we have revised it to ensure greater clarity, coherence and emphasis on the key findings and their clinical relevance.

Reviewer 2 Report

Comments and Suggestions for Authors

The introduction does not sufficiently contextualize how other published multi-parameter or machine learning approaches have succeeded or failed. What specific gaps remain? A more detailed table is needed to perform the literature review.

What is new or improved about your multi-parameter modeling approach compared to existing frameworks?

More details about selection and potential biases (e.g., dropouts, missing visits) are needed for the dataset.

It remains unclear how each scoring system aligns, differs, or complements the other.

Overemphasis on “satisfactory accuracy” without deeper clinical context.

The neural network model performed poorly for DLCO.

The SHAP analysis identifies “top 15” predictors. However, the manuscript does not address correlations among those predictors or discuss whether certain features might be redundant.

With “opacity cutoff: 0.12%” or “high opacity cutoff: 0.002%” appear extremely small. The clinical significance of such minimal changes in lung volume being “abnormal” is unclear—are these thresholds truly meaningful in practice?

The paper concedes “model overfitting” could be an issue. However, it lacks any systematic attempt (e.g., a learning-curve analysis or thorough regularization strategy) to demonstrate that overfitting is minimal or under control.

Author Response

Point 11

Issue

The introduction does not sufficiently contextualize how other published multi-parameter or machine learning approaches have succeeded or failed. What specific gaps remain? A more detailed table is needed to perform the literature review.

Response

We appreciate the reviewers’ comments regarding the Introduction and the need for a more comprehensive contextualization of existing multi-parameter and machine learning approaches. In response, we have revised the Introduction to better address existing multi-parameter models for predicting post-COVID lung function, particularly studies that have used machine learning approaches.

We have opted to summarize the relevant literature within the text rather than using a comparative table, as the studies identified differ substantially, making a direct tabular comparison challenging. Instead, we provide a structured discussion in the Introduction and Discussion, which we believe improves clarity and relevance.

Point 12

Issue

What is new or improved about your multi-parameter modeling approach compared to existing frameworks?

Response

In line with the new literature review section, we have also clearly highlighted the research gaps in this field and outlined how our study addresses these gaps by integrating clinical, laboratory, and CT-derived imaging features into a machine-learning framework.

Point 13

Issue

More details about selection and potential biases (e.g., dropouts, missing visits) are needed for the data set.

Response

We appreciate this point. While the CovILD cohort has been already extensively described in our previous papers (2–5), we provided in the revised manuscript information on the drop-out rates at particular time points as compared with the initial collective of n = 145 patients. We also references numbers and percentages for the categorical modeling responses (DLCO < 80%, FEV1 < 80%, and FVC < 80%).

Point 14

Issue

It remains unclear how each scoring system aligns, differs, or complements the other.

Response

We acknowledge the reviewer’s comment regarding the need for clarity on how the two scoring systems (CT severity scoring (CTSS) and AI-driven lung density quantification) align, differ, and complement each other. To address this, we have clarified the methodology by providing a more detailed description of how CTSS was assessed and how it differs from Syngo Via Pneumonia Analysis. In our data set, CTSS, and software-based quantification (opacity, and high opacity) are strongly inter-correlated.

We have expanded the Methods section to clarify the assessment of both scoring systems and now state explicitly in the Discussion section how both scoring systems contribute to DLCO prediction.

Point 15

Issue

Overemphasis on “satisfactory accuracy” without deeper clinical context.

Response

We apologize for this vague term. In the revised manuscript, we avoid suggestive phrases on model performance and stick to the performance metrics and evaluations.

Point 16

Issue

The neural network model performed poorly for DLCO.

Response

In Review Figure 1 we present correlations between the observed DLCO expressed as percentages of the patient’s reference and the out-of-fold predictions of DLCO in participant-blocked 10-repeats 10-fold cross-validation for all four machine learning algorithm. To facilitate the comparison, we set ranges of the X and Y axes to the range of observed DLCO values - we applied this transformation to the respective scatter plots in Figure 2 as well. In this visualization it is clear that predictions by the neural network algorithm were shrunk to the 75% - 100% range and the neural network algorithm failed to predict cases with very low and very high values of DLCO. We can only speculate on the reasons for this behavior, yet, inclusion of more observations with at the lowest and the highest tail of DLCO would likely performance of not only the neural net but also of the remaining models.

Point 17

Issue

The SHAP analysis identifies “top 15” predictors. However, the manuscript does not address correlations among those predictors or discuss whether certain features might be redundant.

Response

Thanks for this is a very good point. In the initial manuscript, we presented results of exploratory analysis of the modeling data set including the overlap between radiological lung findings, and correlations between opacity, high opacity and CTSS (Supplementary Figure S3) - which were also identified as highly influential explanatory factors for prediction of DLCO < 80% and DLCO (Figure 3 and Supplementary Figure S14). In the revised manuscript, we specifically addressed associations between the most influential explanatory variables with an analysis of a correlation graph (6,7) (Supplementary Figure S15). The graph edges were defined by pairwise correlations between the explanatory factors with Kendall’s  0.3 and weighted by  values. For computation of the correlations, ordinal factors such as presence of GGO or severity of acute COVID-19 were transformed to integers. This analysis supports the findings of the initial manuscript version that the CT readouts, CTSS, opacity, high opacity, GGO and reticulations,  are tightly inter-correlated. Another cluster of tightly associated features was formed by readouts of severity of acute COVID-19 such as acute COVID-19 severity, hospital and ICU stay, and anti-infective treatment during acute COVID-19. Another community of the influential features was made up by known risk factors of severe COVID-19 course (age, cardiovascular disease, hypercholesterolemia, and endocrine/metabolic disease). This leads us to the conclusion that there was indeed some redundancy between the modeling variables and a potential for pruning e.g. by regularization. These new results were also briefly discussed.

Point 18

Issue

With “opacity cutoff: 0.12%” or “high opacity cutoff: 0.002%” appear extremely small. The clinical significance of such minimal changes in lung volume being “abnormal” is unclear—are these thresholds truly meaningful in practice?

Response

We thank the reviewer for this insightful comment. We acknowledge that the functional impact of such minimal lung volume changes is not immediately intuitive, making it difficult to conceptualize how these subtle structural alterations could contribute to pulmonary impairment.

To better illustrate this, we have included a new figure that visually demonstrates a representative case. This example highlights a patient with mild residual CT abnormalities, quantified as 1.86% opacity and 0.06% high opacity, yet presenting with a significantly reduced DLCO of 60%. This supports the notion that even small residual CT changes may have functional consequences, reinforcing the importance of long-term monitoring. Further, we compared our findings to existing literature in the Discussion section.

Point 19

Issue

The paper concedes “model over-fitting” could be an issue. However, it lacks any systematic attempt (e.g., a learning-curve analysis or thorough regularization strategy) to demonstrate that over-fitting is minimal or under control.

Response

Thanks for raising this important issue. As suggested, we tacked it by a learning curve analysis (8) for the predictions of DLCO < 80% and DLCO by the best performing GBMm models presented in Supplementary Figure S13. The over-fitting was particularly evident for the predictions of DLCO < 80% as visualized by a slow convergence of curves of accuracy and Cohen’s  for training and test subsets of increasing sizes derived from the modeling data and cross-validation. By contrast, we can argument that over-fitting of for predictions of DLCO expressed as percentages of the patient’s reference was under control. We list over-fitting explicitly as a limitation in the revised manuscript and propose regularized algorithms such as XGBoost as a possible remedy. We we will certainly consider such algorithms in future studies.

In order to investigate the over-fitting behavior of the models of DLCO < 80%, we resorted to an analysis of the cross-validated model’s performance as a function of increasing DLCO percentages. As shown in Supplementary Figure S12, we found out that the models made uncertain/erroneous predictions - and hence generalized poorly - for observations with DLCO ranging from 70% - 80% of the patient’s reference. This phenomenon can be explained by the observation that the 80% DLCO cutoff used in multiple COVID-19 papers (4,9) did not yielded two clearly separated groups of observations in our modeling data set and may not correspond to substantial differences in demographic and clinical background. In addition, we believe that expansion observations with DLCO < 80%, either be increasing the cohort size or augmentation of the data set, e.g. by SMOTE (10), would at least partly correct this behavior.

Round 2

Reviewer 1 Report

Comments and Suggestions for Authors

Dear Authors, 

Thank you for submitting the revised version and response. After careful review, I have found the paper is still not in the form of acceptance, it is very weak in terms of the write-up and experiment. for example

Concern 1:

Reviewer: The paper lacks details about hyper-parameters for machine learning models. For example, what was the C and class weight value for the SVM model?

Response: Thanks for this point. We agree that the information on hyper-parameters in the initial submission was too specific and hard to understand for many clinical readers. In the revised manuscript, we briefly describe the hyper-parameters in Supplementary Methods/Multi-parameter modeling of lung function. These descriptions were also added as footnotes of Supplementary Table S6.

Comment: Providing only good parameters is not sufficient. There is a need to demonstrate the effect of different parameters so that it can be concluded that your selected value is good.

Concern 2:

Reviewer: Have you considered the GridSearchCV for model hyper-parameters or set it manually?

Response: Thanks for this question. The hyper-parameters were found by searching grids of values with the maximum Youden’s J statistic for the classification models, and minimum mean absolute error (MAE) for the regression models in 10-repeats 10-fold patient-blocked cross-validation as the selection criteria. This tuning procedure was executed by calling train() function from caret package (1), which, together with our open-source package caretExtra, belongs to our standard toolbox for explainable machine learning. This information is provided in Supplementary Methods/Multi-parameter modeling of lung function.

Comment: Have you specifically considered the GridSeachCV (https://scikit-learn.org/stable/modules/generated/sklearn.model_selection.GridSearchCV.html) to find the best parameters? If yes, then which range of values (parameters grid) have you provided to find the best values? 

Furthermore, the details of practical implication and also conclusion are not well written. Please thoroughly improve the paper in terms of practicality and writeup.

Reviewer 2 Report

Comments and Suggestions for Authors

The authors answered all my questions. The manuscript has been sufficiently improved to warrant publication in Diagnostics.

Author Response

(The authors gave the same response as above.)

Round 3

Reviewer 1 Report

Comments and Suggestions for Authors

Dear Authors, 

Thank you for the responses and for updating the manuscript. Some grammar and formatting issues will be fine after the English editing and conversion service of journal.